# Nonmuscle Myosin-2B Regulates Apical Cortical Mechanics, ZO-1 Dynamics and Cell Size in MDCK Epithelial Cells

**DOI:** 10.3390/cells14151138

**Published:** 2025-07-23

**Authors:** Marine Maupérin, Niklas Klatt, Thomas Glandorf, Thomas Di Mattia, Isabelle Méan, Andreas Janshoff, Sandra Citi

**Affiliations:** 1Department of Molecular and Cellular Biology, University of Geneva, 1205 Geneva, Switzerland; marine.mauperin@unige.ch (M.M.);; 2Institute for Physical Chemistry, Georg-August Universität, 37077 Göttingen, Germany; niklas.klatt@uni-goettingen.de (N.K.); thomasjoerg.glandorf@uni-goettingen.de (T.G.);

**Keywords:** nonmuscle myosin-2B, ZO-1, cadherin, actin, cortex, cell size

## Abstract

In epithelial cells, nonmuscle myosin-2B (NM2B) shows a cortical localization and is tethered to tight junctions (TJs) and adherens junctions (AJs) by the junctional adaptor proteins cingulin and paracingulin. MDCK cells knock-out (KO) for cingulin show decreased apical membrane cortex stiffness and decreased TJ membrane tortuosity, and the rescue of these phenotypes requires the myosin-binding region of cingulin. Here, we investigated whether NM2B contributes to these phenotypes independently of cingulin by generating and characterizing clonal lines of MDCK cells KO for NM2B. The loss of NM2B resulted in decreased stiffness and increased fluidity of the apical cortex and reduced accumulation of E-cadherin and phalloidin-labeled actin filaments at junctions but had no significant effect on TJ membrane tortuosity. Fluorescence recovery after photobleaching (FRAP) showed that the KO of NM2B increased the dynamics of the TJ scaffold protein ZO-1, correlating with decreased ZO-1 accumulation at TJs. Finally, the KO of NM2B increased cell size in cells grown both in 2D and 3D but did not alter lumen morphogenesis of cysts. These results extend our understanding of the functions of NM2B by describing its role in the regulation of the mechanical properties of the apical membrane cortex and cell size and validate our model about the role of cingulin–NM2B interaction in the regulation of ZO-1 dynamics.

## 1. Introduction

The actomyosin cytoskeleton is involved in key cellular functions, such as cell division and migration, and controls the shape, remodeling, and mechanical properties of the membrane cortex [1,2,3,4,5]. In epithelial cells the actomyosin cytoskeleton is also critically implicated in the assembly and function of tight junctions (TJs) and adherens junctions (AJs) (reviewed in [6,7,8,9,10,11]). In addition, it regulates the tortuosity of the TJ membrane [12,13,14] and the dynamics of TJ proteins [15,16,17,18]. Epithelial cells express three nonmuscle myosin-2 (NM2) isoforms (NM2A, NM2B, and NM2C), which have different biochemical and contractile properties and functions [4,11,19,20,21,22,23,24,25,26,27,28,29,30] (reviewed in [11,31,32,33,34,35,36]). NM2A and NM2B copolymerize and show a similar distribution in epithelial cells, throughout the membrane cortex and focally accumulated at cell–cell apical junctions [24,37,38,39,40]. However, high-resolution microscopy shows that only NM2B is detected in the juxta-membrane region of AJ during early phases of assembly, where it associates with a network of branched actin filaments [41]. Concerning the specific functions of NM2 isoforms at junctions, in SKCO15 epithelial cells NM2A, but neither NM2B or NM2C, was found to be necessary for the assembly and disassembly of the apical junctional complex (AJC, comprising TJs and AJs) and for TJ barrier function [24,30]. In other studies, depletion of NM2B affected the integrity of E-cadherin and actin filament organization at cell–cell contacts in MCF-7 cells [39] and the organization of the juxta-membrane F-actin meshwork that couples peri-junctional actomyosin bundles to the plasma membrane in MDCK cells [41]. However, whether NM2B contributes to the regulation of the mechanical properties of the apical membrane cortex, TJ protein dynamics, and cell size is not clear.

We recently showed that the TJ protein cingulin and the TJ/AJ protein paracingulin interact directly with NM2B and are required to recruit it to epithelial apical junctions [42]. The knock-out (KO) of cingulin in MDCK cells leads to decreases in ZO-1 accumulation at TJs [43], apical membrane cortex stiffness, TJ membrane tortuosity, and junctional phalloidin labeling [42]. Since these phenotypes could only be rescued by constructs of cingulin that contain the NM2-binding region, and NM2B binds to cingulin with higher affinity than NM2A, we hypothesized that NM2B could be mechanistically involved in these phenotypes independently of cingulin [42,44]. To address this question, we generated and analyzed clonal lines of MDCK cells KO for NM2B. We show here that the loss of NM2B results in decreased tension and stiffness of the apical cortex of MDCK cells, mimicking the KO of cingulin, but did not affect TJ membrane tortuosity. Moreover, the KO of NM2B increased ZO-1 dynamics, correlating with decreased accumulation of ZO-1 at TJs, mimicking a ZO-1 mutant that does not interact with cingulin [43]. Finally, the KO of NM2B increased cell size. These results provide new information about the cellular roles of NM2B.

## 2. Materials and Methods

### 2.1. Experimental Model

MDCK (Madin–Darby Canine Kidney type II cell line, female) cells, originated from the proximal tubule of the kidney, were cultured with Dulbecco’s Modified Eagle’s (DMEM) medium containing 10% Fetal Bovine Serum (FBS), 1% non-essential amino acids (NEAA), 100 units/mL penicillin, and 100 µg/mL streptomycin (P/S) (referred to as standard medium) in a humidified incubator at 37 °C, under 5% of CO_2_ [18]. For AFM experiments, MDCK cells were maintained in minimum essential medium (MEM) containing Earle’s salts, 2 mM GlutaMAX™, 2 g/L NaHCO_3_, and 10% Fetal Bovine Serum Premium (M10F^−^ medium). All commercial materials catalog numbers, manufacturers names and addresses are shown in Supplemetary Appendix A.

### 2.2. Genome Engineering

Clonal lines of NM2B-KO MDCK cells were generated by CRISPR/Cas9 gene editing. Guide RNAs targeted sequences in exons 5 and 7 of the NM2B gene (Appendix A) and were cloned into a CRISPR plasmid comprising GFP for the identification of transfected cells. The plasmids were transfected in wild-type (WT) MDCK II cells using JetOptimus. Single GFP-positive cells were sorted using a Beckman Coulter MoFlo Astrios sorter (Beckman Coulter Life Sciences, Brea, CA, USA) at the Flow Cytometry Platform of Geneva Medical School and seeded into 96-well plates containing an MDCK cell-conditioned medium. Individual clones were expanded and screened for NM2B KO by sequencing, immunofluorescence (IF) microscopy, and immunoblot (IB) analysis. Three NM2B-KO clonal lines were generated (clones #A4, #A6, and #D10), containing either sequence insertions or deletions. γ-actin-KO MDCK cell lines were described previously [18].

### 2.3. Transfection and Exogenous Expression of Proteins and siRNA-Mediated Depletion

For FRAP experiments, MDCK cells were seeded into glass-bottom dishes (27 mm diameter). At 24 h after seeding, cells were transfected with full-length ZO-1 tagged with GFP [43] using the jetOPTIMUS DNA transfection reagent. FRAP imaging was performed 3 days after transfection.

For siRNA-mediated depletion, 1 × 10^5^ cells were seeded on a round glass coverslips. At 24 h after seeding, cells were transfected for 8 h with Lipofectamine RNAiMAX and either siControl, siNM2A, or siNM2B (50 nM, Appendix A) in OptiMeM medium [18]. Cells were fixed for immunofluorescence 3 days after transfection.

### 2.4. Immunofluorescence Microscopy Analysis

For immunofluorescence microscopy analysis, cells were seeded on round glass coverslips in a 24-well plate. Once confluent, cells were fixed in 1% PFA at room temperature (RT) for 7 min, followed by methanol fixation at −20 °C for 5 min and gradual rehydration in PBS. Permeabilization was performed with 0.2% TritonX-100/PBS for 4 min at RT, and cells were saturated in 2% BSA/PBS for 30 min at RT. Primary antibodies (Appendix A) were incubated for 3 h at RT, followed by an incubation with DAPI and secondary antibodies (Appendix A) for 2 h at RT. Coverslips were mounted using Fluoromount-G. Samples were imaged using a Zeiss LSM800 confocal microscope equipped with a Plan-Apochromat 63×/1.40 oil objective. Images were acquired at 1024 × 1024 px resolution and processed as previously described [18].

The TJ membrane tortuosity (zigzag index) was calculated as the ratio (L(TJ)/L(St)) between the total length of the bicellular junction (L(TJ)), measured using the freehand line tool in Fiji/ImageJ, and the straight-line distance between two vertices (L(St)), measured with the straight-line tool in ImageJ [12]. A total of 60 junctions were typically analyzed for each experimental condition.

To quantify junctional immunofluorescence, the absolute fluorescence intensity (AFI), referring to signal pixel intensity (mean gray value) for the related channel, was measured in the selected junctional area using the polyhedral tool of ImageJ. A total of 120–160 junctions were typically analyzed for each experimental condition.

To quantify the cell size, cell areas were measured using a junctional marker and the freehand line tool in ImageJ. A total of 70–100 cells were analyzed.

### 2.5. Atomic Force Microscopy (AFM)-Based Force Measurements

Indentation relaxation measurements were performed using a NanoWizard4 (JPK Instruments, Berlin, Germany) mounted on an inverted microscope (IX 81; Olympus, Tokyo, Japan). Silicon nitride cantilevers with a nominal spring constant of 0.01 N m^−1^ (MLCT C) were used. The sensitivity of the AFM was determined by recording force curves in a µ-dish (35 mm, ibiTreat) filled with M10F^−^ medium without cells, and the exact spring constant for each cantilever was determined using the thermal noise method [45].

To measure the viscoelastic properties of the apical cell cortex, 2 × 10^5^ cells were seeded in the inner ring of an ibiTreat dish and grown to confluence. The cells were washed twice with M10F^−^ containing 0.2 mg/mL penicillin, 0.2 mg/mL streptomycin, and 15 mM HEPES, referred to as M10F^+^. The samples were mounted on the AFM stage, and 2 mL of M10F^+^ medium was added. The Petri dish heater (JPK Instruments, Berlin, Germany) was set to maintain the temperature at 37 °C.

Indentation was performed at a constant speed of 2 µm/s until a maximum force of 1 nN was reached, corresponding to an indentation depth of approximately 1 µm. At this point the feedback was switched to a constant distance, and the force relaxation was monitored over time. After a dwell time of 0.5 s at constant height, the cantilever was retracted at the same speed. Five consecutive force curves were recorded at the apical center of individual cells within the monolayer using the same indentation parameters. Force curve analysis was performed as previously described [46], and unsuitable fits were filtered out using a custom-written Python (version 3.10, Python Software Foundation, Wilmington, DE, USA) script. This tension-based model accounts for both the viscoelastic properties of the cortex–membrane interface and the cortical tension generated by actomyosin contractility [18].

### 2.6. Fluorescence Recovery After Photobleaching (FRAP) Experiments

MDCK cells transiently expressing ZO-1 tagged with GFP were cultured in 27 mm glass-based dishes for 72 h. FRAP experiments were conducted in HBSS supplemented with 15 mM Hepes pH 7.4 at 37 °C on a Leica Stellaris8 confocal microscope (Leica Microsystems, Wetzlar, Germany) using a 63x immersion objective. The photobleaching was performed on cell junctions four times during 1:280 s at 70% laser power, and fluorescence recovery was monitored for 10 min post-bleaching. Image analysis involved calculating the fluorescence intensity (F) at the junctional (Fj) and background (Fb) regions, corrected for photobleaching in the control region (r): F = (Fj − Fb)/r. The mobile fraction and half-time (t1/2) of fluorescence recovery were then determined using a one-phase decay nonlinear regression fit in the GraphPad Prism software (Boston, MA, USA, version 10) [18].

### 2.7. Immunoblot Analysis

Cell lysates were prepared using RIPA buffer (150 mM NaCl, 40 mM Tris-HCl, 1% Triton X-100, 10% glycerol, 2 mM EDTA, 0.2% SDS, 0.5% deoxycholate, pH: 7.5) containing fresh PIC (ThermoScientific, #A32965), and IB was performed as described previously [18]. Cell lysates were sonicated (8 s at 66% amplitude using a Branson sonicator (Emerson, St. Louis, MO, USA)), and denatured at 95 °C for 5 min. Protein samples (10 µg) were separated on 8% to 15% polyacrylamide gels and transferred to 0.22 µm nitrocellulose membranes (Roth, #9302-1) at 90 V for 1.30 h at 4 °C. Membranes were incubated with primary antibodies (Appendix A) overnight at 4 °C, followed by a 1 h incubation at RT with secondary antibodies (Appendix A). Protein detection was carried out using the WesternBright ECL kit (Advansta, #K12045-D50) and the Amersham ImageQuant 800 (Cytiva) imaging system. Signal intensity of bands was quantified using Fiji/ImageJ software. The relative signal intensity was calculated as the ratio of the protein of interest to the β-tubulin reference standard.

### 2.8. Lumen Morphogenesis

To induce cyst formation, 40 µL of Matrigel (BD Biosciences, #354230) was spread onto a glass coverslip in a 24-well plate and allowed to solidify for 30 min at 37 °C. Cells were trypsinized, centrifuged (1500× *g* for 3 min), and resuspended in S-MEM to obtain a single-cell suspension. A total of 3.5 × 10^4^ cells were mixed 1:1 with a solution containing 2× standard DMEM medium, 4% Matrigel, and 10 ng/mL Epidermal Growth Factor (EGF). Next, 400 µL of this solution were plated onto the solidified Matrigel. On day 7, cysts were fixed in methanol/acetone (1:1) for 11 min at −20 °C, permeabilized in 0.5% Triton X-100/PBS for 10 min at RT, and blocked with IF buffer (PBS with 100 mM glycine, 0.5% BSA, 0.2% Triton X-100, and 0.05% Tween-20) for 2 h at RT. Primary antibodies were incubated overnight at RT, followed by three washes with IF buffer. Secondary antibodies and DAPI were applied for 4 h at RT, followed by three washes with IF buffer. Coverslips were mounted with Fluoromount-G. Samples were imaged using a Zeiss LSM800 confocal microscope(Carl Zeiss Microscopy, Jena, Germany) and a Plan-Apochromat 63×/1.40 oil objective, at 1024 × 1024 px resolution. Images were processed using ImageJ (.czi extraction) and assembled in Affinity Designer. For quantifications, 60 cysts were analyzed. Cysts were classified based on the presence and number of lumens (no lumens, one, two, or multiple lumen(s)) [44], diameter, and height.

### 2.9. Quantification and Statistical Analysis

Data processing and analysis were performed using the GraphPad Prism8 software (Version 10). All experiments were conducted in duplicate or triplicate. Results are displayed as dot-plots, histograms, or line-graphs with mean values and standard deviations provided to illustrate variability. Statistical significance was assessed using either a two-sided one-way Anova test, preceded by checking the normal distribution using the Kolmogorov–Smirnov test, or a two-sided unpaired Mann–Whitney test (ns: not significant, * *p* < 0.0332, ** *p* < 0.0021, *** *p* < 0.0002, **** *p* < 0.0001).

## 3. Results

### 3.1. The Knock-Out of NM2B Results in Decreased Phalloidin and E-Cadherin Junctional Labeling

We obtained distinct clonal lines of MDCKII cells (termed MDCK hereafter) knock-out (KO) for NM2B using CRISPR/Cas9 gene editing technology. No expression of NM2B was detected in the KO cells based on immunofluorescence microscopy (IF) of mixed cultures of WT and NM2B-KO cells (Appendix A) and immunoblot (IB) analysis of cell lysates (Appendix A). The levels of expression of the two other NM2 isoforms (NM2A and NM2C) in the three NM2B-KO clonal lines were similar to WT cells, based on IB analysis (Appendix A, quantification on the right), in agreement with previous results on epithelial cells depleted of NM2B through siRNA [24,39,41,47].

Studies on MCF7 cells showed that depletion of NM2B resulted in a decrease in the intensity of phalloidin staining in the apical junctional ring and affected the integrity of cadherin organization at cell–cell contacts [39]. IF microscopy analysis of NM2B-KO MDCK cells confirmed that both phalloidin labeling (Figure 1A, top panel, quantifications on the right) and E-cadherin labeling (Appendix A, quantification on the right) were slightly but significantly decreased in NM2B-KO cells compared to WT cells. On the other hand, IB analysis did not reveal altered levels of E-cadherin (Appendix A, quantification on the right), suggesting that the decrease in junctional labeling for cadherin is due to altered assembly/retention at junctions and not to decreased expression. To extend this analysis and ask whether the decreased phalloidin labeling correlated with changes in actin isoform accumulation at junctions, we used antibodies specific for either β-actin or γ-actin [48]. IF analysis of β-actin (Figure 1A, middle panel, quantifications on the right) and γ-actin (Figure 1A, bottom panel, quantifications on the right) showed no difference in the junctional labeling for the two isoforms when comparing neighboring KO and WT cells. Moreover, immunoblot analysis showed no difference in the expression levels of pan-actin, β-actin, and γ-actin in the three different clonal lines of NM2B-KO cells with respect to WT (Figure 1B, quantification below the immunoblots). Together, these findings suggest that NM2B maintains the tensile state of junctional actin filaments in a manner similar to cingulin, without affecting total actin levels and actin isoform composition [42]. In addition, the KO of NM2B also affects the junctional accumulation of E-cadherin, unlike the KO of cingulin.

### 3.2. The Knock-Out of NM2B Reduces Apical Membrane Cortex Stiffness but Does Not Perturb TJ Membrane Tortuosity

Next, to determine the potential role of NM2B in the decreased apical stiffness of cingulin-KO cells [42], we examined the mechanical properties of the apical cortex of NM2B-KO cells. Analysis by atomic force microscopy (AFM) using conventional sharp tips showed that apical cortical tension was not impaired by the KO of NM2B (Figure 2A, Table 1). In contrast, cortical stiffness (area compressibility modulus, $K_{A}^{0}$) was significantly decreased in NM2B-KO cells compared to WT cells (Figure 2B, Table 1). In addition, the fluidity (β) of the cortex was significantly increased in NM2B-KO cells compared to WT cells (Figure 2C, Table 1).

Second, we analyzed TJ membrane tortuosity, as measured by the zigzag index [12], and found that it was similar in NM2B-KO and WT cells (Figure 2D, quantification on the right), in agreement with previous observations on MDCK cells depleted of NM2B through siRNA [18]. In summary, the KO of NM2B, similarly to the KO of cingulin, causes a decrease in apical stiffness, but unlike the KO of cingulin, it has no effect on TJ membrane tortuosity.

### 3.3. The KO of NM2B Promotes Increased ZO-1 Exchange and Decreased Accumulation of ZO-1 at TJs

ZO-1 exchanges between junction-associated and cytoplasmic pools, and its dynamic behavior depends on NM2 activity and ZO-1 binding to actin [15,16]. In addition, a mutant of ZO-1 that lacks the cingulin-binding C-terminal ZU5 domain is more dynamic and shows decreased accumulation at TJs [43], raising the hypothesis that cingulin stabilizes ZO-1 by tethering it to NM2B [42]. To test this hypothesis, we used fluorescence recovery after photobleaching (FRAP) of exogenously expressed GFP-tagged ZO-1 in MDCK cells. The KO of NM2B significantly accelerated FRAP recovery of GFP-ZO-1 when compared to the recovery in WT cells (Figure 3A). ZO-1 showed a significantly higher mobile fraction in NM2B-KO cells (75% in NM2B-KO versus 62% in WT) (Figure 3B) and a tendency toward lower half-times (57% in NM2B-KO versus 63% in WT cells) (Figure 3C, respective kymographs in Figure 3D, live imaging in Appendix A). This indicates that the KO of NM2B results in more rapidly exchangeable ZO-1.

To determine whether this phenotype correlated with altered ZO-1 accumulation at TJs, we carried out IF microscopy analysis. The KO of NM2B resulted in a slight but significant decrease in the junctional accumulation of ZO-1 compared to WT cells (Figure 3E, top panel, quantification on the right), while protein expression levels of ZO-1, cingulin, and occludin were not affected (Figure 3F, quantification below the immunoblots). In contrast to ZO-1, the junctional accumulation of both cingulin and occludin was not affected by the KO of NM2B (Figure 3E, middle and bottom panels, respectively, quantification on the right). Together, these results suggest NM2B stabilizes ZO-1 at TJs.

### 3.4. The KO of Either NM2B or γ-Actin Increases Cell Size in Cells Grown as Monolayers in 2D or Cysts in 3D

In MDCK cells there is a correlation between apical stiffness and cell size [49], and previous studies in vivo show increased size of cardiac myocytes in NM2B-KO mice [50] and increased size of γ-actin-depleted cells [51]. However, whether either NM2B or γ-actin regulates the size of MDCK epithelial cells is not known. To address this question, we compared the cell area of WT/NM2B-KO and WT/γ-actin-KO cells grown either in 2D as monolayers or 3D as cysts. Cell area was significantly increased in NM2B-KO cells compared to WT cells (142.2 μm^2^ for NM2B-KO versus 119.2 μm^2^ for WT) (Figure 4A, quantification on the right). A similar increase in cell area was observed following siRNA-mediated depletion of NM2B (189 μm^2^ for siNM2B versus 127.4 μm^2^ for siControl) (Figure 4B, quantification on the right). In contrast, depletion of NM2A did not alter cell size (119.2 μm^2^ for siNM2A, 113.4 μm^2^ for siControl) (Figure 4C, quantification on the right). Cell size was also increased in γ-actin-KO cells (112.8 μm^2^ for WT, 206.2 μm^2^ for γ-actin-KO) (Figure 4D, quantification on the right). Both NM2B-KO and γ-actin-KO cells formed well-structured cysts with mostly one lumen, similar to WT cysts (Figure 4E,F and Figure 4I,J respectively, quantifications on the right), indicating that NM2B and γ-actin are not required for normal lumen morphogenesis in vitro. However, the cyst diameter was significantly larger in both NM2B-KO cysts (39.0 μm) (Figure 4G) and γ-actin-KO cysts (35.3 μm) (Figure 4K) compared to WT (33.0 μm). In addition, the thickness of the cysts was also increased (NM2B-KO: 7.15 µm, γ-actin-KO: 6.56 µm, WT: 5.92 µm) (Figure 4H,L). Together, these results indicate that the KO of either NM2B or γ-actin leads to an increase in cell size of MDCK grown either in 2D or 3D without impairing cyst lumen morphogenesis.

## 4. Discussion

Here we provide new insights about the functions of NM2B by showing that it regulates the mechanical properties of the membrane cortex, the dynamics and junctional stabilization of ZO-1, and cell size. Specifically, the KO of NM2B in MDCK cells reduces the stiffness of the apical membrane cortex while increasing its fluidity, increases the exchange of ZO-1 while decreasing its accumulation at junctions, and increases cell size.

In previous studies, the knock-down (KD) of NM2B was shown to inhibit the continuous distribution of E-cadherin and the accumulation of junctional phalloidin-labeled actin filaments [39]. Here we confirm that NM2B maintains normal phalloidin and cadherin labeling at junctions using a different model, e.g., NM2B-KO MDCK cells. Since the labeling for actin isoforms was similar in WT and NM2B-KO cells, and the KO of γ-actin does not result in altered phalloidin labeling [18], these results suggest that NM2B is uniquely required to maintain the tensile state of junction-associated actin filaments. This is in agreement with kinetic and biochemical studies indicating that NM2B is more adapted than NM2A to maintain a high static tension of actomyosin in cells [29,52] (reviewed in [11,31,32,33,34,35,36]). Moreover, our observations indicate that the decreased phalloidin labeling observed in cingulin-KO cells depends on its binding to NM2B, as previously hypothesized [42]. Our results also extend previous observations [39,41] by showing that besides E-cadherin, ZO-1 junctional labeling is also decreased in NM2B-KO cells, correlating with the increased exchange and higher mobile fraction of ZO-1. Since inhibition of NM2A activity by blebbistatin reduces ZO-1 exchange [16,18,53], the increased exchange of ZO-1 in NM2B-KO cells could be due to the replacement of NM2B by NM2A, which, as noted above, is the fastest isoform and more biochemically suited for contractility and rapid turnover. It is also possible that the decreased E-cadherin and ZO-1 junctional labeling implies that they are mechano-regulated by NM2B, through maintenance of actin filament tension, and this mechano-regulation is required for their efficient junctional assembly. ZO-1 is an actin-binding and mechano-sensing protein that exists in extended (open) and folded (closed) conformations [54], and both force and binding to partners such as ZO-2 and cingulin promote its extended conformation in vitro and in vivo, promoting its junctional assembly [42,43,54,55,56]. The observation that NM2B promotes ZO-1 accumulation at junctions independently of cingulin supports a model whereby TJ assembly of ZO-1 depends on the integrity of the cingulin–NM2B module.

One important objective of our study was to determine the role of NM2B in the mechanical properties of the apical membrane cortex, with the goal of understanding its relevance in the mechanism through which cingulin promotes apical cortical stiffness [42]. The phenotype of NM2B-KO cells, with decreased apical membrane cortex stiffness and increased fluidity, was similar to that observed in cells KO for either cingulin or γ-actin [18,42]. This suggests that cingulin, which is required for the junctional tethering of NM2B and for the TJ proximity of γ-actin [42], regulates the apical membrane through its ability to control both NM2B and γ-actin, and that these latter regulate the mechanical properties of the cortex independently of cingulin. Both γ-actin [48] and NM2B [24,39] are in fact localized in the apical cortex, and micro-rheology and confocal imaging show that γ-actin forms stiffer networks [57], accounting for its ability to maintain a stiff apical cortex [18]. In the case of NM2B, its ability to be attached to actin for a longer fraction of the enzymatic cycle than NM2A may contribute to maintaining a stiffer cortex. Another possibility is that the closer spatial association of NM2B with the membrane [41] implies a deeper penetration into the actin cortex, which affects cortical mechanics [58]. In contrast, cingulin is not localized at the apical cortex, with the exception of early embryos [59]. Thus, circumferential TJ anchoring of NM2B and γ-actin by cingulin is necessary to maintain a stiffer apical cortex, in agreement with the idea that the apical cytoskeleton should be considered as a combined structure in association with the TJ [60].

A second mechanical phenotype of CGN-KO cells was reduced tortuosity of the TJ membrane [42]. Since the KO of NM2B did not affect TJ tortuosity, our previous hypothesis that NM2B is uniquely required for the transmission of orthogonal forces [13] from the peri-junctional actomyosin belt to the cingulin-ZO-1 complex is not validated [42]. We postulate that in NM2B-KO cells, NM2A, which also binds to cingulin [42,61], can substitute NM2B as a component of the ZO-1–cingulin–NM2 module. Increased activity and levels of expression of junctional NM2A promote TJ membrane tortuosity [12,14,18,39,62], but the levels and junctional accumulation of NM2A were not increased in NM2B-KO cells, consistent with no change in TJ membrane tortuosity. Since in cingulin-KO cells there is no significant change in NM2A levels, the decreased TJ membrane tortuosity of cingulin-KO cells suggests a major architectural role of cingulin in connecting TJs to the actomyosin cytoskeleton. In summary, our results show that KO of NM2B impacts apical membrane cortex stiffness without affecting TJ membrane tortuosity and confirm that these two mechanical phenotypes are independent.

We report here the novel finding that epithelial cells KO for NM2B show a larger size when grown either in 2D or 3D. This is consistent with the observation that mouse-derived cardiac myocytes show increased size in vivo [50]. Previous studies on γ-actin-depleted fibroblasts and SK-CO15 cells also show increased cell area [51,63], although in the study on SKCO-15 cells [63], the increase in size was reported, but no measurement was provided. Here we confirm a role for γ-actin in regulating cell size, using MDCK cells, and we provide quantitative data. However, the mechanisms underlying these phenotypes are unclear. Cell size is controlled by a combination of internal and external factors, including the rate of biosynthesis and division, cell growth and proliferation, surface-to-volume ratio, and the interplay of various signaling pathways (reviewed in [64,65,66,67,68]. γ-actin depletion impairs cell growth and cell cycle progression in cancer cells [69] and affects the actin cortical network during mitosis and cytokinesis [70], providing potential mechanisms. However, cell shape and size are also determined by the mechanical properties of the cortex [71], and there is a relationship between cell size and cortical stiffness both in developing embryos [72] and MDCK cells [49]. Cells with identical mechanical properties but a larger footprint (i.e., increased radius) tend to appear softer under mechanical probing, particularly at higher strains. This apparent softening arises not from a change in intrinsic mechanical properties but from geometric effects—specifically, the increased surface area over which force is distributed [72]. In our experiments, we observed an approximate 8% increase in cell radius, which could contribute modestly to a reduction in apparent stiffness. However, this geometric effect alone does not fully account for the pronounced softening observed in AFM measurements. Thus, the increase in cell size supports the conclusion that NM2B-KO cells exhibit a genuine reduction in cortical stiffness compared to wild-type controls. How the cell-specific balance in the expression of actin and NM2 isoforms, their localization, and their specific biochemical and mechanical properties influence cell size should be investigated further in future studies.

The physiological relevance of our in vitro findings can be framed in the context of the known roles of NM2B in vivo. In mice, NM2B is required for the development of both heart and brain, and its KO causes embryonic lethality due to hydrocephalus, linked to abnormalities in the cell adhesive properties of neuroepithelial cells [21,32,50,73,74]. We speculate that altered cadherin and ZO-1 assembly at junctions could affect cell adhesion during development, and future studies should assess whether the composition and function of cardiac and neuroepithelial cell junctions are altered upon KO of NM2B in mice. Similarly, it is not clear whether the NM2B-dependent effects on the mechanical properties of the apical membrane cortex of cultured epithelial cells occur in vivo either in the developing embryo or in adult tissues. One interesting model is the cochlear epithelium, where mutations of both cingulin and γ-actin result in altered tissue architecture and cell loss in vivo, leading to deafness [75,76,77,78]. NM2B is also expressed in the cochlear epithelium, and its mutation in mice alters the extension, patterning, and morphology of the cochlear inner pillar cells [47]. Future studies should therefore also characterize the effects of KO or mutation of NM2B on junction composition, apical cortex mechanics, and auditory function.

## 5. Conclusions

We provide new evidence that NM2B plays non-redundant roles in the maintenance of apical membrane cortex mechanics, the tensile state of junctional actin filaments, the junctional assembly and stabilization of ZO-1, and cell size in cultured MDCK epithelial cells.

## Figures and Tables

**Figure 1 cells-14-01138-f001:**
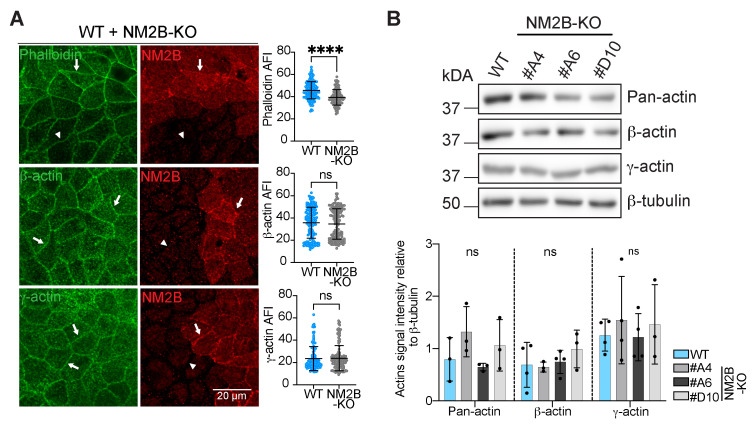
The KO of NM2B decreases the junctional accumulation of phalloidin labeling without affecting individual actin isoforms. (**A**) Immunofluorescence (IF) microscopy analysis and absolute fluorescence intensity (AFI) quantification (on the right) of endogenous phalloidin-labeled F-actin (top panels), β-actin (middle panels), and γ-actin (bottom panels) (green) at junctions in mixed cultures of WT and NM2B-KO. Cells were labeled with antibodies against NM2B to distinguish WT from NM2B KO cells (red). Arrows indicate normal labeling for phalloidin, β-actin, and γ-actin (as in WT cells), and an arrowhead indicates loss of NM2B or phalloidin labeling in KO cells. The scale bar corresponds to 20 µm. Statistical significance of quantitative data was determined by a two-sided unpaired Mann–Whitney test (ns: not significant, **** *p*-value < 0.0001) (N = 2, n = 120–160 junctions). (**B**) Immunoblot (IB) analysis and relative densitometric quantifications (on the bottom) of the protein levels of pan-actin, β-actin, and γ-actin in lysates of WT or NM2B-KO MDCK cells (3 distinct clonal lines). β-tubulin was used as a loading control. Dots show replicates (N = 3–4), and bars represent mean ± SD. Statistical significance of quantitative data was determined by a two-sided one-way Anova test (ns: not significant).

**Figure 2 cells-14-01138-f002:**
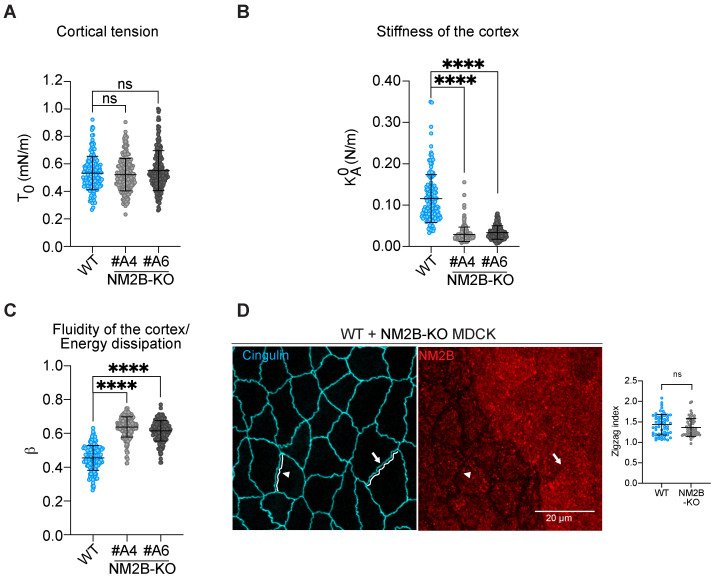
The KO of NM2B decreases apical cortical stiffness while increasing its fluidity but does not perturb cortical tension and TJ membrane tortuosity. (**A**–**C**) Tension (T_0_) (**A**), stiffness *(*$K_{A}^{0}$*)* (**B**), and fluidity (β) (**C**) of WT and NM2B-KO MDCK cortex (n = 151 for WT, n = 223–224 for NM2B-KO #A4, and n = 170–181 for NM2B-KO #A6). Statistical significance of quantitative data was obtained using a two-sided one-way Anova test (ns: not significant, **** *p*-value < 0.0001). (**D**) IF microscopy analysis and zigzag index quantifications (on the right) of endogenous cingulin (cyan), a marker of TJ, in co-culture of WT and NM2B-KO cells. The white lines show traces used for measurement of TJ membrane tortuosity. Dots show replicates (N = 2, n = 60 junctions), and bars represent mean ± SD. Statistical significance of quantitative data was obtained using a two-sided unpaired Mann–Whitney test (ns: not significant).

**Figure 3 cells-14-01138-f003:**
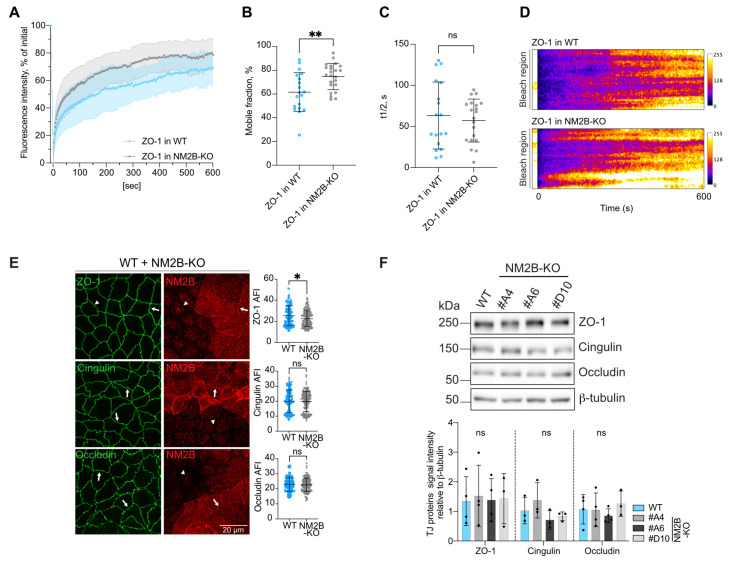
The KO of NM2B promotes increased ZO-1 exchange and reduces junctional ZO-1. (**A**) Quantitative FRAP analysis of overexpressed ZO-1 tagged with GFP in WT (blue dots) or in NM2B-KO (grey dots) MDCK cells. (**B**,**C**) Mobile fraction (**B**) and half-time (t1/2) values (**C**) of exogenous GFPZO-1 tagged with GFP determined from FRAP curves shown in (**A**). Statistical significance of quantitative data was obtained using a two-sided unpaired Mann–Whitney test (ns: not significant, ** *p*-value < 0.0021) (N = 3, n = 18–23 junctions). (**D**) Representative kymographs of exogenous GFP-tagged ZO-1 either in WT (top panels) or in NM2B-KO (bottom panels) MDCK cells. Related Appendix A show FRAP live imaging of ZO-1 tagged with GFP in WT (S1) and in NM2B-KO (S2) MDCK cells. (**E**) IF microscopy analysis and absolute fluorescence intensity (AFI) quantification (on the right) of the localization of ZO-1, cingulin, and occludin (TJ markers, green) in co-cultures of WT and NM2B-KO cells, with WT cells identified by NM2B labeling (red). Arrows indicate normal labeling for TJ markers (as in WT cells), and an arrowhead indicates loss of NM2B or decreased ZO-1 labeling in KO cells. The scale bar corresponds to 20 µm. Statistical significance of quantitative data was obtained using a two-sided unpaired Mann–Whitney test (ns: not significant, * *p*-value < 0.0332) (N = 2, n = 154–160 junctions). (**F**) IB analysis and relative densitometric quantifications (on the bottom) of ZO-1, cingulin, and occludin in lysates of WT or NM2B-KO MDCK cells (3 distinct clonal lines). β-tubulin was used as a loading control. Dots show replicates (N = 3–4), and bars represent mean ± SD. Statistical significance of quantitative data was obtained using a two-sided one-way Anova test (ns: not significant).

**Figure 4 cells-14-01138-f004:**
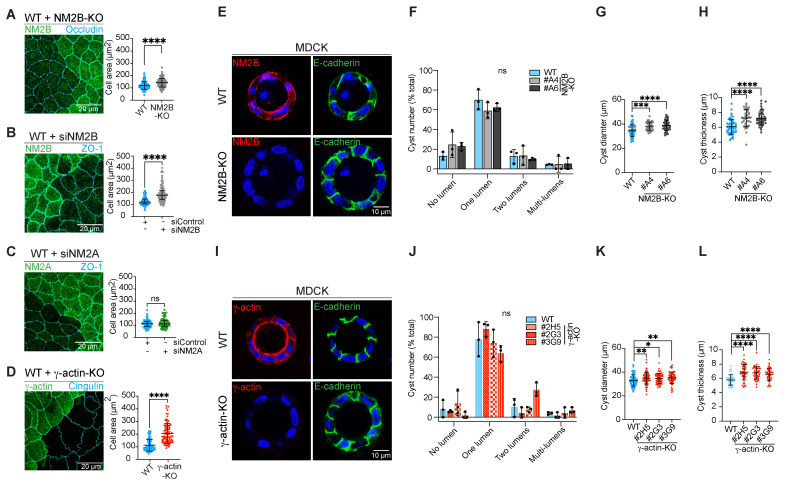
Effects of the KO of NM2B on cell size and lumen morphogenesis in MDCK cells. (**A**) IF microscopy analysis and cell area (quantification on the right) in co-cultures of WT (n = 133) and NM2B-KO (n = 133) MDCK cells, identified by labeling with NM2B (green), from two independent experiments. Endogenous occludin (cyan) was used to label individual cells as a reference. (**B**,**C**) IF microscopy analysis and cell area quantification (on the right) of endogenous ZO-1 (cyan), used to label individual cells, in WT cells treated with siNM2B (n = 88–89) (**B**) or siNM2A (n = 73) (**C**), identified by labeling with antibodies against NM2B or NM2A (green), from three independent experiments. (**D**) IF microscopy analysis and cell area quantification (on the right) of endogenous cingulin (cyan), used to label individual cells in co-cultures of WT (n = 86) and γ-actin-KO (n = 88) MDCK cells, identified by labeling γ-actin (green), from three independent experiments. (**E**–**I**) IF microscopy analysis of endogenous NM2B or γ-actin (red) and E-cadherin (green) in WT and NM2B-KO (**E**) or γ-actin-KO (**I**) MDCK cells using the 3D cyst model. The scale bar corresponds to 10 μm. (**F**,**J**) Quantifications of the number of cysts with no lumen, or containing one, two, or multiple lumen(s) in WT (n = 50–59) and NM2B-KO (2 distinct clonal lines, n = 42–43) (**F**) or γ-actin-KO (3 distinct clonal lines, n = 58–81) cysts. (**J**) MDCK cells from three independent experiments. Dots show replicates, and bars represent mean ± SD. Statistical significance of quantitative data was determined by a one-way Anova test (ns: not significant). (**G**–**K**) Quantifications of the cyst diameter (μm) in WT (n = 59) and NM2B-KO (2 distinct clonal lines, n = 42–43) (**G**) or γ-actin-KO (3 distinct clonal lines, n = 58–81) (**K**) MDCK cells, from three independent experiments. (**H**–**L**) Quantifications of the cyst thickness (μm) in WT (n = 46) and NM2B-KO (2 distinct clonal lines, n = 37–43 (**H**) or γ-actin-KO (3 distinct clonal lines, n = 45–47) (**L**) MDCK cells, from three independent experiments. (A–D, G–H, K–L) The scale bar corresponds to 20 μm. Dots show replicates, and bars represent mean ± SD. Statistical significance of quantitative data was determined by a two-sided unpaired Mann–Whitney test (ns: not significant, * *p*-value < 0.0332, ** *p*-value < 0.0021, *** *p*-value < 0.0002, **** *p*-value < 0.0001).

**Table 1 cells-14-01138-t001:** Viscoelastic parameters (mean ± SD) from indentation curve fitting in WT and NM2B-KO MDCK cells (Figure 2A–C).

Genotype, Clone	T0 (mN/m)	$K_{A}^{0}$ (N/m)	β
WT	0.534 ± 0.122	0.116 ± 0.058	0.455 ± 0.073
NM2B-KO A4	0.522 ± 0.118	0.029 ± 0.017	0.638 ± 0.059
NM2B-KO A6	0.551 ± 0.146	0.034 ± 0.016	0.616 ± 0.060

## Data Availability

Data generated in this study will be made available in the Yareta repository upon acceptance and will be provided upon reasonable request.

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
