# Peer review of "Nonmuscle Myosin-2B Regulates Apical Cortical Mechanics, ZO-1 Dynamics and Cell Size in MDCK Epithelial Cells"

_cells, 2025, doi:10.3390/cells14151138_

Round 1
Reviewer 1 Report
Comments and Suggestions for Authors
This manuscript deals with the role of Non Muscle Myosin IIB in the formation, stabilization and maintenance of Tigth junctions complexes and in particular ZO1. It is meant to test the hypothesis made by the authors of a joined role of cingulin/paracingulin and NMHHB in apical cortex stiffness and TJ stability.
The authors show that the KO of NMIIB in MDCK cells partially mimicks cingulin KO. The only phenotype that is not phenocopied is TJ tortuosity.
The work is well done, the experiments are clearly described and the data support conclusions. However, the work is rather incremental. The only very novel finding is the absence of effects on TJ tortuosity that authors did not investigate further.
The reviewers noticed also a few points that need to be clarified:
The authors do not give any valid explanation on the fact that phalloidin staining is decreased at junctions while beta and gamma actin are not.
The values for parameters of AFM experiments should be compared with those obtained previously by the authors on cingulin and paracingulin KO.
FRAP experiments: authors state in the text that half-times are lower for KO than for WT, while the difference is not given as statistically significant on Figure 3.
The quality of images of panel 3E does not allow to assess the claims of the authors on ZO1, cingulin and occludin accumulations at junctions.
line 340 the statement "inhibition of NMIIA activity by blebbistatin" is not appropriate as blebbistatin inhibits the three isoforms.
lines 353-358 are very unclear as line 328-331. It is difficult from the very verbose discussion to catch the mechanisms proposed by the authors and its novelty in regard to previous data. In general this discussion, that deals with very litle new/unexpected data should be reduced by more than half.
Author Response
The replies are in the attached document.

Reviewer 2 Report
Comments and Suggestions for Authors
This is an informative paper that reveals new information into the role of non-muscle Myosin IIB in tight junction mechanics and molecular dynamics. Earlier, the authors demonstrated that the TJ protein, cingulin, was necessary for junctional tension and mechanics. There, the Myosin II-binding domain seemed to play an important role based on structure-function analysis of domain deletion mutants of cingulin.
The current paper directly tests this inference using independent NMIIB knock-out cell lines. It clearly shows that NMIIB supports TJ molecular stability and tension, and also contributes to cortical stiffness and elasticity. The data quality is excellent and convincing.
Author Response
The replies are in the attached document

Reviewer 3 Report
Comments and Suggestions for Authors
The manuscript by Mauperin et al, is an analysis of NM2B in MDCK cells using knockout models on its role in regulation of cell-cell junctional proteins, actin organization, cortex stiffness and membrane tortuosity. The novelty of the work is not clear, as lot of data on the role of NM2B exists in regulation of junctional proteins. Can the authors further highlight this in abstract, discussion?
Using phalloidin and actin staining, authors conclude that KO of NM2B decreases junctional accumulation of F-actin. This does not look appropriate conclusion. According to the presented data, it would be safer to say there is no change in F-actin localization or expression.
Have authors tried to perform any of the assays in more physiological conditions e.g. in primary epithelial cells? Have authors tried any assays in serum starvation conditions? Is there any phenotypic differences between control and NM2B ko cells? What is the outcome of changes in stiffness of the cortex in terms of physiological relevance? Can any assays be performed? Can the phenotype be rescued with overexpression of NM2B in ko?
Author Response

(The authors gave the same response as above.)

Reviewer 4 Report
Comments and Suggestions for Authors
In their study, Maupérin and colleagues used a non-muscle-myosin 2B (NM2B) knockout (KO) model to analyse the involvement of this protein in apical membrane stiffness and the tortuosity of tight junctions (TJ), as observed in cingulin knockout. In their MDCK II KO model, they observe decreased stiffness and tension, as well as effects on E-cadherin and ZO-1.
While the study is interesting and employs various sophisticated techniques, there are several points that need to be considered.
- Major points:
1. It is unclear why two clones were used in some experiments and only one or pooled data in others. For cortical tension (Fig. 2A), one clone is not different from the wild type, but the other is. Therefore, it can be concluded that cortical tension is not altered due to the loss of NM2B, but rather due to other factors.
In general, it is suggested that experiments be performed on more than two clones, especially when the two clones show differences.
2. The differences in junctional accumulation (Fig. 1A) are quite low, although significant. Is there a functional effect to be expected?
3. Similarly to 2., the effects seen on ZO-1 are quite small compared to the effects seen on stiffness. Are there other factors involved that were not studied, or how could such a small change lead to such a significant effect?
4. In the discussion, the authors report that NMB2 is important for heart and brain development and is also essential for neuroepithelial cells. It would be worthwhile studying the effect of NMB2 KO in relevant cell models. Perhaps kidney epithelial cells are not the optimal model? Furthermore, it is worth considering whether effects might be more relevant or different in a tight epithelial cell model. MDCK I cells could be of interest, especially when auditory function depends on NMB2, as a tight epithelium is usually necessary in this case. Have the authors considered the functional consequences, such as changes in transepithelial electrical resistance?
- Minor points:
1. In the final version, ensure that higher-resolution versions of some figures are submitted. For example, the resolution in Figures 1 and 3 could be improved.
2. Table 1: Please provide n-numbers and resulting p-values.
3. In the 'Materials and Methods', under '2.1 Genome engineering'. 'MDCK WT' refers to different cells to 'MDCK II'. Please change this to 'wild-type MDCK II' or similar.
4. Materials and Methods > 2.9 Quantification and statistical analysis. The definition of the limits for the p-values is rather uncommon. This does not need to be changed and should be taken more as a comment.
5. Please provide a larger image of Figure 2D; the staining is difficult to evaluate without magnifying it significantly.
6. In Figure 3F, it would be better to show a blot where there is no air bubble directly in the band of interest.
Author Response

(The authors gave the same response as above.)

Round 2
Reviewer 3 Report
Comments and Suggestions for Authors
the authors updated the manuscript.
Author Response
Thank you for your positive evaluation of the revision
Reviewer 4 Report
Comments and Suggestions for Authors
The revised manuscript of Maupérin and colleagues deals with the raised issues and furthermore contains additional new data improving the whole study.
Below the reply to the authors’ response can be found with an additional comment at the end about the new data.
In short, the manuscript may be accepted, when the authors replace figures by their new versions. They seem to have forgotten this for figure 2 and 3, as here the described changes can not be found in the new version of the manuscript.
Major points:
- It is unclear why two clones were used in some experiments and only one or pooled data in others. For cortical tension (Fig. 2A), one clone is not different from the wild type, but the other is. therefore, it can be concluded that cortical tension is not altered due to the loss of NM2B, but rather due to other factors. In general, it is suggested that experiments be performed on more than two clones, especially when the two clones show differences.
Response: We did have three clones at the beginning of our study, and we carried out some experiments (immunoblot analyses) using all the three clones, which gave essentially identical results. However, we later lost one of the three clones due to the failure of the nitrogen tank where this clon was stored. This is why some experiments, including the biophysical experiments, were carried out with only 2 clones. In the field of biophysics, the standard is to use only one representative clone. Using two clones, we repeated the experiments additional times, which allowed us to provide updated, more reliable values for tension. We revised both text and figures accordingly.
Reply: The changes are acceptable. However, as a comment, I still wonder how one can decide which clone is representative when only two clones were analyzed ion detail and are differing in findings.
- The differences in junctional accumulation (Fig. 1A) are quite low, although significant. Is there a functional effect to be expected?
Response. The Reviewer is correct, the differences are small but significant, similarly to what bserved in Smutny et al (Nat. Cell Biol., 2010). In the text of Results and Discussion, we revised the text to underline this fact.
Reply: The revision is okay.
- Similarly to 2., the effects seen on ZO-1 are quite small compared to the effects seen on stiffness. Are there other factors involved that were not studied, or how could such a small change lead to such a significant effect?
Response: We analyzed approximately 150 junctions to assess the effect of NM2B-KO on ZO-1 junctional localization. The difference reached statistical significance (p = 0.016, unpaired Mann–Whitney test), corresponding to p < 0.0332 (*). The Reviewer is correct that this is less significant than the differences observed for cortical stiffness and fluidity (p < 0.0001, ****), but the two phenotypes cannot necessarily be compared directly. KO of cingulin also decreases the junctional accumulation of ZO-1 in a small but significant manner, which we believe reflects the fact that cingulin binds to NM2B with higher affinity than NM2A.
Reply: Okay.
- In the discussion, the authors report that NMB2 is important for heart and brain development and is also essential for neuroepithelial cells. It would be worthwhile studying the effect of NMB2 KO in relevant cell models. Perhaps kidney epithelial cells are not the optimal model? Furthermore, it is worth considering whether effects might be more relevant or different in a tight epithelial cell model. MDCK I cells could be of interest, especially when auditory function depends on NMB2, as a tight epithelium is usually necessary in this case. Have the authors considered the functional consequences, such as changes in transepithelial electrical resistance?
Response: The Reviewer is correct that each epithelial cell type is different, and results may vary between epithelial cell lines. The reason why we used MDCKII cells (see also Response to Rev. 3) is that all our previous characterization of CGN-KO, CGNL1-KO and -actin-KO cells was carried out using MDCKII cells. For consistency, we had to use the same model. We hope that our results will encourage more researchers to test the function of NM2B in additional models in vitro and in vivo, because each epithelial cell type is distinct in their expression of junctional proteins (see our 2019 paper Vasileva et al, where we compare the expression levels of a series of junctional proteins in several types of epithelial and endothelial cells in culture). The TEER values at steady state depend mainly on the expression of claudin isoforms, and not on the composition of the NM2A/NM2B/NM2C actomyosin cytoskeleton. Indeed, NM2B depletion by siRNA did not impair TEER in SKCO-15 cells (Ivanov et al., 2007). Based on this, we did not prioritize TEER measurements in our experiments.
Reply: Okay.
- Minor points:
- In the final version, ensure that higher-resolution versions of some figures are submitted. For example, the resolution in Figures 1 and 3 could be improved.
Response: We improved the quality of the figures to increase their resolution.
Reply: For Figure 1 this change was done. But Figure 3 seems still to be the same – see also below.
- Table 1: Please provide n-numbers and resulting p-values.
Response: These information are provided into the Source Data files.
Reply: Okay.
- In the 'Materials and Methods', under '2.1 Genome engineering'. 'MDCK WT' refers to different cells to 'MDCK II'. Please change this to 'wild-type MDCK II' or similar.
Response: The text was revised as required: ‘The plasmids were transfected in wild-type (WT) MDCK II cells using JetOptimus (Polyplus, #101000051).’
Reply: Ok.
- Please provide a larger image of Figure 2D; the staining is difficult to evaluate without magnifying it significantly.
Response: The size of this image was increased as required.
Reply: By comparing it with the old version, no increase of the image can be found.
- In Figure 3F, it would be better to show a blot where there is no air bubble directly in the band of interest.
Response: A new blot without air bubble in the band of interest was provided.
Reply: Figure 3 still is the same as before. The author may have forgotten to replace the figures within the manuscript.
Comments to the new data:
- Please also increase the quality of Figure 4 as well.
Author Response
Comment: The changes are acceptable. However, as a comment, I still wonder how one can decide which clone is representative when only two clones were analyzed ion detail and are differing in findings.
Response: in our experience two clones are sufficient and the differences in their behaviour using different assays are negligible.
Comment: For Figure 1 this change was done. But Figure 3 seems still to be the same – see also below.
Response. In Fig. 3 Panel F the image of the blot of ZO-1 was changed in the first revision, there is no air bubble.
Comment (R1) Please provide a larger image of Figure 2D; the staining is difficult to evaluate without magnifying it significantly.
Response: The size of this image was increased as required.
Reply (R2): By comparing it with the old version, no increase of the image can be found.
Response (R2): we further increased the size to 35 mm each side of the squares of the images, we cannot go higher without enlarging significantly the whole Figure, which would mean adding another line for the quantification plot. This Figure represents a negative result (no change in TJ membrane tortuosity), so we feel that what is important is not the large size of the image but especially the quantification of TJ tortuosity (histogram on the right).
Comment (R1) 6. In Figure 3F, it would be better to show a blot where there is no air bubble directly in the band of interest.
Response (R1): A new blot without air bubble in the band of interest was provided.
Reply: Figure 3 still is the same as before. The author may have forgotten to replace the figures within the manuscript.
Response (R2) we will re-upload the modified Figure, according to our record the revised Figure did contain the image with no air bubble.
Comments to the new data:
- Please also increase the quality of Figure 4 as well
Response: Unfortunately this Figure is very dense with data, and we cannot increase the size of the immunofluorescence images without decreasing the size of the other panels, which would make them unreadable. We think that quality of the data is adequate to convey the scientific message. In panels A-D one can see clearly enough the differences in cell area between WT and KO cells, and in panels E and I the difference in cyst size and thickness between WT and KO are also clearly visible. The labelling of NM2B, gamma-actin and E-cadherin are there only to confirm the genotype and show cell outlines.